# Machine Learning Methods in Clinical Flow Cytometry

**DOI:** 10.3390/cancers17030483

**Published:** 2025-02-01

**Authors:** Nicholas C. Spies, Alexandra Rangel, Paul English, Muir Morrison, Brendan O’Fallon, David P. Ng

**Affiliations:** 1Department of Pathology, University of Utah, Salt Lake City, UT 84112, USA; 2ARUP Laboratories, Division of Applied Artificial Intelligence, Institute for Research and Innovation, Salt Lake City, UT 84108, USAbrendan.o’

**Keywords:** clinical flow cytometry, acute leukemia, machine learning, operational efficiency

## Abstract

Machine learning or artificial intelligence can be used to effectively analyze flow cytometry data from hematologic malignancies. Here we review three main methods in machine learning: supervised, unsupervised, and weakly supervised learning and how they can be applied to clinical flow cytometry. Additionally, we review the operational, logistical, and regulatory considerations that underlie a clinical implementation of these tools.

## 1. Introduction

The quality and quantity of data generated with each passing year are increasing exponentially due to the emergence of new testing methodologies and the rapid decrease in testing cost as existing technologies mature. Within the field of flow cytometry, this is most apparent in the steady march towards higher color flow cytometers including 3 color systems in 1991 [1], 17 color systems in the early 2000’s [2], spectral cytometers in the early 2010s [3], and now 40+ color panels [4], coupled with the exponential increase in events collected per sample [5] and the increase in the number of samples being run both clinically [6] and in research [7]. Traditional methods of manual interpretation and gating are unable to cope with the magnitude of data generated and, thus, over the past few decades, numerous attempts have been made towards applying machine learning methods in analyzing flow cytometry data [8,9]. The history and an extensive review of these efforts is beyond the scope of this paper; however, we will endeavor to provide a basic introduction to machine learning as it pertains to flow cytometric analysis including an overview of computational and bioinformatics methods, a review of techniques used in model development and training, and a primer on the practical aspects of implementation in the laboratory setting.

Among laboratory test methodologists, flow cytometry lends itself to computational methods as the outputs from machines are inherently digital, as opposed to other methods such as FISH (Fluorescence In Situ Hybridization) or gel electrophoresis, which generate images requiring subsequent digitization to apply computational methods. In brief, single fluorescently labeled cells are interrogated by exciting lasers, and the Stokes-shifted photons are filtered into narrow bands and detected by individual detectors (photomultiplier tubes [PMTs] or avalanche photodiodes [APDs]). This results in a voltage output that is digitized via an analog-to-digital converter for each cell/event on each detector, resulting in a data table with N events and M detectors. While this data table can be stored in any format, the need to preserve additional information (also known as *metadata*) including date and time, instrumentation parameters, users, fluorophore and antibody information, and spillover values has motivated the field to gravitate towards the flow cytometry standard (FCS), which was first introduced in 1990 [10] and has since undergone multiple revisions, most recently in 2021 [11]. In the most modern implementations, these data are stored in a raw format without scaling transformations and compensation for spectral spillover effects with the understanding that these transformations and compensation will be applied after data collection during the time of data analysis. As of 2024, several bioinformatics packages written in Python and R have been developed to access this data format, including FlowKit/FlowIO [12] (https://github.com/whitews/flowkit, accessed on 10 October 2024) for Python and flowCore [13] for R, among others. These data are converted into arrays, spectral compensation is applied, and then various transformations (e.g., logarithmic, arcsinh, or logicle) could be applied to make visualization and clustering easier. Of note, spectral compensation is simply the matrix inversion of the spillover table stored in the metadata to recover as much of the original concentration of fluorophore/antibody constructs as possible. Further data quality and filter checks are often performed in an automated fashion including detection for fluidic issues on the time gate, doublet exclusion, viability gating, and debris exclusion [14,15] before the data are passed on to various machine learning pipelines. Once these preanalytical steps are complete, much of the flow cytometry subject matter expertise gives way to classical machine learning on tabular data, as we will see in later sections.

## 2. A Primer on Machine Learning for the Flow Cytometrist

Machine learning (ML) is a discipline within artificial intelligence that focuses on developing computational algorithms that can (1) simulate the performance of a human task and (2) improve their performance on said task without requiring explicit instructions on how to do so. Machine learning systems typically incorporate inputs from multidimensional datasets to identify patterns or make inferences about how those inputs ultimately relate to a desired output. In the context of flow cytometry, machine learning algorithms can incorporate the digital signals being measured from each cell and color to provide predictions or insights on any potential pathophysiology that may be present in the sample being analyzed. The volume, complexity, and annotations typical of flow cytometry data make it an ideal application for machine learning solutions. As such, an understanding of the foundational principles of machine learning is essential for pathologists and flow cytometrists seeking to integrate these computational tools into their workflows.

One of the primary tasks of machine learning is to identify patterns or make predictions based on input data. This process typically involves several stages including data input, feature extraction, model training, and model validation. Flow cytometry data, composed of millions of events each measured across multiple parameters (such as fluorescence intensities from different channels), serve as the input. From these raw data, meaningful features are extracted and refined, after which a portion of the data is used to train the machine learning model. This model is tasked with adjusting its internal parameters to accurately represent the relationship between the input features and the desired outcomes. After the training phase, the model is validated on a separate dataset to evaluate its performance and ensure it can generalize to new unseen data.

Classically, machine learning methods are described in relation to the proportion of their training data that is accompanied by a ground truth label, referred to as degrees of “supervision”. In brief, supervised learning refers to tasks in which the entire training data set is annotated with an output or “ground truth” label (see Section 3 below). In flow cytometry, these labels are typically the presence or absence of disease states, such as acute myeloid leukemia. These labels are often procured using subject matter experts, such as a pathologist’s final diagnosis from routine clinical care. Algorithms such as logistic regression, support vector machines, and neural networks are common supervised learning methods used to classify cells or disease states. They are often quite highly performant on the tasks for which they are trained, but only if the input training data and their respective labels are of sufficient volume and quality. The phrase “garbage in, garbage out” often comes to mind when troubleshooting performance for these supervised classifiers.

In contrast, unsupervised learning (Section 4) deals with datasets that lack explicit labels or predefined outcomes. This is particularly relevant in flow cytometry for identifying and characterizing novel or rare cell populations within complex high-dimensional data. Unsupervised learning algorithms, like k-means clustering, hierarchical clustering, and more advanced methods such as FlowSOM [16], UMAP [17] (Uniform Manifold Approximation and Projection), and t-SNE (t-Distributed Stochastic Neighbor Embedding), can group similar cells together based on their measured parameters without any prior labeling. These algorithms help uncover underlying structures in the data, potentially revealing previously unrecognized cell populations or identifying unique patterns of expression that may have diagnostic significance. These approaches are often most useful when generating hypotheses, or when the process of acquiring high-fidelity reference labels at scale is unfeasible.

Between supervised and unsupervised methods lies a set of newer paradigms, such as self-supervised, semi-supervised, and weakly supervised approaches (see Section 5). Each of these methods aims to leverage a partial set of labels that are made available or infer them from the existing structure within datasets.

In real-world applications for machine learning problems, the utility of a developed model is closely linked with its ability to make inferences on data that it has not seen before. The ability to predict the true labels for new data points with similar accuracy as those on which it was trained is referred to as “generalizability”. At the heart of the issue of generalizability is another fundamental concept in machine learning, the bias–variance trade-off. The bias–variance trade-off refers to striking a balance between *bias*, the error introduced by simplifying complex relationships too much, and *variance*, the error introduced by excessive sensitivity to fluctuations in the training data. Achieving an optimal balance ensures that a model generalizes well to new unseen data by being neither overly simplistic (known as underfitting) nor excessively complex (known as overfitting). High bias results in models may consistently miss relevant patterns, for example, collapsing to a single pattern found in annotated labels and ignoring rare labels, while high variance leads to models tailored too closely to the training data and unable to perform accurately on testing data. Avoiding under- and overfitting is a crucial consideration throughout the model development process, with implications for training data inclusion/exclusion or transformation, model architecture selection, and hyperparameter tuning, among others. Techniques such as cross-validation, regularization, early stopping, and careful model selection are often employed to mitigate these risks, ensuring the model performs robustly across different datasets.

To properly evaluate the performance and generalizability of machine learning models, we have a plethora of useful tools at our disposal. These include metrics well known to the practicing laboratorian, such as accuracy, sensitivity, specificity, positive predictive value, and negative predictive value. However, there are a few key idiosyncrasies in machine learning parlance that are worthy of explicit mention. First, the issue of class imbalance. A common task in machine learning applications is the identification of rare events, where positive cases outnumber negatives by orders of magnitude. Where typical laboratory assays may be evaluated by their sensitivity and specificity across a range of thresholds using a receiver operating characteristic curve, machine learning solutions are often evaluated by their trade-off between sensitivity and positive predictive value. These metrics are more commonly referred to in the ML literature as *recall* (sensitivity) and *precision* (PPV) and are combined on a precision–recall (PR) curve. Metrics such as the Matthews correlation coefficient (MCC) [18] and the F1 score are helpful single measures of performance that are more common in ML applications, especially those with significant class imbalance.

Overall, machine learning offers powerful methods for analyzing complex flow cytometry data. By learning to classify cell populations, identifying unseen patterns, and improving the accuracy and efficiency of data analysis, these algorithms can significantly aid pathologists. Integrating these principles and tools into the laboratory workflow requires a collaborative effort. Pathologists, data scientists, and laboratory professionals must work together to ensure that the models developed are trained on high-quality data and are properly validated. With this foundation, pathologists can confidently interpret machine learning outputs, evaluate the performance of these models, and harness their full potential to enhance diagnostic precision and deepen our understanding of cellular biology through flow cytometry. The remainder of this review aims to provide these fundamentals.

## 3. Supervised Machine Learning Methods

Supervised machine learning encompasses a huge diversity of algorithms and approaches, but the key unifying aspect is the use of *annotated* data, i.e., data examples which have been assigned a ground truth label via manual review (see Figure 1). In other words, whatever we want our ML model to predict, whether that is a discrete classification of samples (e.g., presence of absence of a disease state) or a continuously valued regression target (e.g., what fraction of cells in a sample are of a certain type), supervised methods require a human expert to provide those target values for each example in the training set. Once trained, the ML model can then make similar predictions on previously unseen data.

In a medical setting, the need for annotations is both the greatest strength and weakness of supervised methods: while the provided annotations in a sense directly encode the knowledge of domain experts and allow the model to learn from this expertise, generating these annotations requires a potentially large expenditure of time and effort from those same domain experts. Removing this need for expert annotations is a major motivation behind many unsupervised and self-supervised machine learning approaches (discussed in subsequent sections).

Practitioners must beware that most supervised algorithms, if left to their own devices, will *overfit* to their provided training data, such that the model’s performance metrics as computed on the training data will not be at all representative of their future performance on unseen data. It is standard practice, therefore, to split the available annotated data into *training* and *validation* sets. The model is never trained on the validation set; instead, only inference is run on the validation set and performance metrics computed, which (it is hoped) are more representative of the model’s performance on future data.

Common practice is to randomly place 80% of available data in the training set and reserve 20% for validation, but exceptions and subtleties abound. For example, if the distribution of data over time is known or suspected to be variable (e.g., drift in lab instruments, lot-to-lot variability in reagents, etc.), it may be advantageous to use only the most recent 20% of data for validation so as to not dangerously overestimate future performance. Or, for a multi-class classification problem, it is often useful to ensure that the relative frequencies of different classes are approximately the same in training and validation sets, especially if these frequencies are highly imbalanced. Furthermore, many algorithms have auxiliary “hyperparameters” that must be tuned for optimal performance, and this performance can only be measured by running inference on data not included in the training set. But, any validation data used for this tuning are, in a sense, “contaminated” and no longer suitable as an honest measure of future performance. In this case, it is common to use a *train*/*validation*/*test* split, where the model is trained on only the training data, inference over the validation data is used to select optimal hyperparameters for the model, and then final performance is judged on the test set using the optimally tuned model. (Note that the terminology is not standardized and in many studies the role of validation and test splits are often reversed from the description here.)

### 3.1. Survey of Commonly Used Supervised ML Methods in Flow Cytometry

A brief summary of the algorithms to be discussed in the rest of this section can be found in Table 1. Before considering specific algorithms, it is worth noting a subtlety for the common task of the sample-level classification (instead of cell-level classification) of flow cytometry data. Most of the supervised learning methods discussed below cannot use the raw cell/event data directly and must first transform, aggregate, or otherwise dimensionally reduce the raw data to sample-level *features*, and it is these sample-level features which the supervised classifier takes as input. The reduction to sample-level features is often achieved using an *un*supervised method such as a self-organizing map (SOMs), Gaussian mixture models (GMMs), Uniform Manifold Approximation and Projection (UMAP), or one of many other choices. Some of these representative unsupervised methods are discussed below in Section 4.

### 3.2. Support Vector Machines (SVMs)

A basic SVM classifier can be formulated as the problem of finding a decision boundary that separates labeled examples in the training set. In essence, data samples are viewed as vectors in some high-dimensional space, and the algorithm attempts to find a hyperplane in this vector space which separates labeled examples by their respective class. So-called kernel methods generalize this idea by first transforming the data to a higher-dimensional vector space with a chosen non-linear mapping (the kernel), and then solving for the optimal hyperplane in this transformed space. This substantially improves the representational power of the model since, while the decision boundary is necessarily linear in the transformed space, the decision boundary in the original space can be quite non-linear. The optimization problem is solved subject to a loss function which is a sum of two terms, one which seeks the smoothness of the decision boundary (which should help generalization) and the other which seeks to minimize the misclassification of training examples (since an excessively smooth boundary may miss important features of the data). Minimizing the total loss function reflects a tradeoff between the competing demands of these smoothness and accuracy terms, with their relative importance usually controlled by a tunable hyperparameter. Though most easily conceptualized for binary classification tasks, SVMs can also be generalized to multi-class problems and even regression tasks. For more details on SVMs, we recommend the recent review by Guido et al. [19]. By way of example in the flow cytometry domain, in 2015, the University of Washington group [20] transformed raw cytometry event data into 2D histograms of all pairwise combinations of markers (visually similar to the dot plots commonly used in a manual review of cytometry data, except without gating), and then applied an SVM classifier to the resulting intensity maps. Similarly, the AHEAD group instead fit Gaussian mixture models to the raw cytometry data to generate specimen-level representations, and then used those representations as input to an SVM classifier [21].

### 3.3. Ensemble Techniques: Decision Trees, Random Forests, and Gradient-Boosted Trees

Typical decision tree classifiers construct a binary tree in an appealingly intuitive fashion: choose a feature in the dataset (which can be numerical or categorical), search for a splitting of the dataset on the feature which is at least weakly predictive of the desired class label, and recursively extend the tree in the same manner. There are many details which we ignore here, such as how to choose features for splitting, what stopping criteria to use for the recursion, et cetera, which were recently reviewed by Meinye et al. [22]. Despite their intuitive simplicity, plain decision tree classifiers are rarely used in machine learning as they tend to wildly overfit the training data and generalize quite poorly to new data. But, they form the basis for other algorithms, which are widely used and more robust.

If a single decision tree is suboptimal, imagine instead that we construct an *ensemble* of many different decision trees from the training data and classify based on a vote of the trees. To generate diversity in the ensemble, individual trees can be trained on only a subset of the dataset and its features. This is the basic idea of a *random forest* [23]. Surprisingly, such an ensemble of individually weak classifiers can itself be a very strong classifier. A powerful refinement of this idea is *gradient-boosted trees*, where one constructs the ensemble sequentially, attempting to minimize the error of the existing ensemble with each new added tree, instead of constructing trees independently of each other, as in a random forest. Gradient-boosted tree models, such as the widely used implementation provided by XGBoost [24], are the de facto standard for supervised ML on tabular datasets and commonly outperform most other approaches (with the possible exception of neural networks; see below). In our experience, these two techniques are fast, computationally inexpensive (or embarrassingly parallelizable on modern computing hardware), and challenging to outperform when dealing with dataset sizes seen within the clinical arena (in the order of 10^2^ and 10^5^ training examples as a gross generalization). As a whole, they are not as prone to overfitting and have less sensitivity to initialization and hyperparameters compared to other classification techniques such as neural networks. As examples, random forest classifiers on UMAP (see the Unsupervised Learning section) projections of raw flow cytometry data saw use in classifying various B-cell malignancies [25]. Additionally, random forests were leveraged in identifying Hodgkin lymphoma from a relatively limited dataset [20]. Similarly, our group has used an XGBoost classifier on top of SOM [26] projections to classify acute myeloid leukemia cases; this is noteworthy as not just a proof-of-concept study, but a fully deployed model in clinical use.

### 3.4. Neural Networks (NNs)

At their core, most modern neural networks (NNs) are “just” gigantic stacks of individually simple arithmetic operations. Trainable weight matrices multiply input data and pass the results through dozens, even hundreds, of successive layers of cleverly arranged matrix multiplications and simple non-linear functions. With appropriate training, the net effect can be an almost arbitrarily complex mathematical function.

NNs have exploded in the popular imagination in recent years and are now the undisputed champions of natural language processing and computer vision. Nevertheless, for tabular data problems such as flow cytometry, the landscape is not as simple. The performance of gradient-boosted trees and NNs are typically comparable, and only rarely does one outperform the other by large margins. [27]

Many recent works have successfully applied NNs to flow cytometry problems. Simonson et al. [28] transformed raw flow cytometry data to 2D histograms and used convolutional NNs on these images to detect Hodgkin lymphoma. Similarly, others have used SOMs [29,30] (discussed below) to aggregate cell data into specimen-level representations, which were fed to NN classifiers to discriminate various B-cell neoplasms. Avoiding the unsupervised aggregation of most methods described so far, Hu et al. [31] applied NNs directly to mass cytometry data, combined with a simple pooling operation and further NN layers, for the specimen-level detection of cytomegalovirus infections. And, in a break from specimen-level classification, the Mayo group [32] invested the effort to generate *cell*-level annotations on approximately 200 patient samples and used this to train an NN *cell* classifier for the minimal residual disease testing of chronic lymphocytic leukemia.

### 3.5. Multiple Instance Learning (MIL)

Most of the supervised ML methods described thus far require an *un*supervised method to “featurize” the raw cell data when applied to specimen-level flow cytometry tasks, which adds complexity. Multiple instance learning (MIL) methods neatly sidestep this necessity. In this framework, a specimen—a *bag* in the jargon—is modeled as a permutation-invariant collection of *instances*, i.e., individual cells, and it is assumed during training that we have annotations at the bag level but no annotations of individual instances. The ML task is then to predict bag-level labels. Ilse et al. [33] represents a typical modern approach, which first uses an NN block to separately transform each instance into some latent representation, a second NN block to aggregate the instance representations into a single bag representation, and a final NN block to classify the bag representation. Importantly, the aggregation NN block is designed to learn which instances are or are not predictive of the bag label and weight them appropriately in the aggregation. The entire sequence of NN layers, though having logically distinct functions, is really one connected NN, which is conveniently trainable as a single model, with the full array of modern NN architectures and training techniques available. Though this pattern of instance encoding, bag pooling, and bag classification is common for MIL problems, alternate approaches [34] are possible, which directly construct dissimilarity measures on entire bags instead of representing bags as aggregations of instance representations. An MIL model along the lines of Ilse et al. [33] was used in Lewis et al. [35] for the detection of acute leukemias from flow cytometry data. In particular, Lewis and coworkers were able to reliably discriminate acute myeloid leukemia from B- and T-lymphoblastic leukemias and also predict the presence or absence of certain cytogenetic aberrancies and genetic variants from flow cytometry data alone.

Besides Lewis et al. [35], we are not aware of any other works applying MIL methods to flow cytometry, but we believe this is a very promising approach for future work. By combining featurization and classification into a single end-to-end trainable model, the entire model development process is significantly simplified. The main challenge for applying MIL to flow cytometry is designing good instance encoders. This is in contrast to Ilse et al. [33] and similar works, which apply MIL to computer vision problems and can therefore leverage powerful off-the-shelf computer vision models as instance encoders, whereas, for flow cytometry, it may not be possible to create such pre-trained general purpose models, and one must instead start from scratch.

## 4. Unsupervised Machine Learning Methods

Unsupervised learning is a branch of machine learning that utilizes unlabeled input data (i.e., unannotated) and leverages the algorithm to elucidate patterns and structure without any prior mappings to outputs (see Figure 2). In contrast to supervised learning methods, there is no requirement for data annotation, which can be both time-consuming and expensive. Additionally, the ability to discover new relationships without a priori knowledge remains a significant merit of these methods in research and exploratory applications. In the context of clinical flow cytometry, these methods offer the potential to identify rare cell populations and novel marker expression profiles. Despite these advantages, the performance of these algorithms can be sensitive to hyperparameter selection, leading to variability and suboptimal performance. These methods suffer from a lack of interpretability due to the nature of the unlabeled input, often requiring domain expertise and supplemental analyses to make meaningful biological associations. Further, applications of these methods on large datasets with millions of cells over thousands of samples can require significant computational resources. We provide an overview of the most common unsupervised methods, as well as recent demonstrations in the field of flow cytometry.

### 4.1. Clustering Algorithms

**K-means**. K-means clustering is the simplest example of an unsupervised learning method, where the data are separated based on feature similarity. The number of clusters, “*k*”, is set during the initialization of the algorithm and defines the number of initially randomly dispersed centroids, which are then optimized to dictate the center of the clusters in a Voronoi partitioned feature space. The simplicity of k-means makes it a popular choice for high-throughput analyses of flow cytometry data, such as the flowPeaks package in R [36]. Freecyto, a web-based application, utilizes weighted k-means clustering to enable the flexible analysis of large datasets, enabling a suite of interactive visualization tools [37].

**Density-Based Spatial Clustering of Applications with Noise (DBSCAN).** DBSCAN is an alternative algorithm which clusters data points based on their density in the input space. High-density regions of points with similar features within a specified radius are grouped together, whereas sparse or low-density regions are designated outliers. This method allows for more flexibility when compared to k-means, with the ability to identify clusters of arbitrary shapes and sizes without preset parameters. FlowGrid [38], enhances the scalability of density-based clustering by partitioning the feature space into a grid-based structure, which reduces the computational complexity. This approach improves the efficiency for large flow cytometry datasets, enabling the clustering of millions of cells in seconds compared to minutes with other methods.

**Gaussian mixture modeling (GMM).** GMM is a probabilistic model that represents data as a mixture of normal distributions to approximate uncertainty in the data. In contrast to other methods, GMM is a soft clustering method that assigns the probability of data points belonging to a certain group, enabling the handling of more complex associations and relationships. GMM has been widely applied in flow cytometry to improve the interpretation of complex cellular patterns. A cytometric fingerprinting strategy, PhenoGMM, was presented to characterize microbial populations [39]. This method automates the gating of bacterial subpopulations, enabling robust quantitative composition analysis across samples. Tailor, a phenotype-aware algorithm, overcomes the noise in varied low-resolution flow cytometry panel modes. [40] This is accomplished by first binning cells based on their marker expression distributions to reduce heavy tail distributions that can mask important cellular information, and subsequently using these binned inputs for GMM clustering. This algorithm was utilized to evaluate a 16-color panel for T cell differentiation and was able to distinguish subtle differences in CD27 expression, which typically appear as unimodal in other analyses. Finally, the AHEAD group applied this technique to initial feature extraction in their publications [21,41] followed by fisher vectorization (another dimensionality reduction technique) and a support vector machine classifier. This technique appears robust; however, in our experience, clustering techniques (be they k-means, DBSCAN, or GMM) suffer from sensitivity to initialization, data drift issues, and hyperparameter sensitivity and, in the setting of non-ideal inter-cytometer and temporal homogenization, these techniques see decreases in performance.

We note that the performance of clustering methods in flow cytometry can vary greatly depending on how similarity between cells is measured. Calculating feature similarity is heavily influenced by the distance metric chosen and how each parameter (e.g., intensity, scatter) is weighted, which can dramatically change clustering results [42]. For instance, Euclidian distance exhibits high granularity and is ideal when absolute marker intensities are imperative, while cosine similarity prioritizes the orientation of data points and is adept at capturing relative expression patterns to distinguish phenotypes. [43] Further, transformations (i.e., log, arcsinh) and scaling methods, commonly employed to optimize performance across a wide dynamic range, can modify the data distribution, thereby altering the distance between points and biasing the definition of each subpopulation. Further, cell populations tend to overlap and may not fall into discrete clusters in a biological context [44]. Consequently, there is no gold standard for defining populations in unsupervised methods and gating strategies can vary between experts, instruments, or experiments. Accordingly, the best approach varies by dataset and objective, where empirical testing is needed to determine the appropriate distance metric to apply.

### 4.2. Linear Dimensionality Reduction Techniques

**Principal Component Analysis (PCA).** PCA is a popular method for linear dimensionality reduction that aims to simplify the feature space by identifying directions or “principal components” that maximize the variance in the data. This technique captures independent variables to extract the most important features from the larger dataset, rendering data easier to visualize. PCA has been a fundamental method in dimensionality reduction and remains a prevalent tool for visualizing multiplexed flow cytometry data. Gachon et al. recently utilized PCA to enhance the interpretability of minimal residual disease (MRD) levels in patients with AML [45]. In this demonstration, PCA enables the standardization of flow cytometry data from multiple patients, enabling cohort-level analysis. PCA revealed that high MRD follow-ups were correlated to AML diagnostic samples, while negative MRD follow-ups were correlated with normal bone marrow, suggesting clinical relevance in the embedding. This technique has significant clinical importance in that it was popularized by the Euroflow consortium under the automatic population separator (APS) moniker and used within their proprietary software Infinicyt (BD Biosciences) [46].

**Non-Negative Matrix Factorization (NMF).** NMF is a method that transforms the dataset into a low-rank approximation of the input feature space. This method transforms a non-negative data matrix into two lower-dimensional basis and coefficient matrices. The algorithm iteratively changes the values in the two matrices so that their product converges to the original input matrix. A notable advancement is the development of NMF-RI, a blind spectral unmixing method optimized for highly mixed multispectral flow cytometry data [47]. This approach enhances the accuracy of fluorescence signal separation without requiring knowledge of individual fluorophore spectra upon initialization, thereby improving the analysis of complex samples.

### 4.3. Non-Linear Dimensionality Reduction Techniques

**Self-Organizing Map (SOM).** An SOM is a neural network algorithm method that simplifies the feature space by representing the input data as a grid of nodes that retains spatial relationships between points. Each neuron in an SOM is assigned a weight vector upon initialization and competes for representation, and the weight that is most similar to a randomly selected vector from the training set, in addition to those nodes in close proximity, is rewarded to reinforce relationships of nearby neurons. Iterative cycles of this process allow the map to grow in different shapes, grouping similar inputs and informing patterns in the data. FlowSOM [16] is a recent key advancement that utilizes self-organizing maps for the scalable analysis of high-dimensional flow cytometry data by clustering cells based on them identifying rare cell populations. The FlowSOM method utilizes a minimal spanning tree to connect nodes of the SOM based on similarity distances, creating a formation that minimizes the distance between nodes. The tree-like structure aids in conceptualizing relationships between the clusters. Subsequently, meta-clustering is used to further refine groupings into biologically meaningful associations. This is achieved through the hierarchical clustering of the nodes of the MST, which enables the elucidation of cell populations and subtypes. While this method has been successfully applied in several projects and appears generally robust to flow cytometry data, like all clustering type techniques, it is still sensitive to initial starting conditions, data quality, and hyperparameters. We have recently demonstrated an SOM-based model that utilizes flow cytometry data to predict AML [26] and the clinical implementation of this algorithm for triaging positive samples for further analysis. We note that this is the first report of a deployed algorithm in a flow cytometry laboratory and highlights the operational advantages of integrating these tools into the clinic.

**Uniform Manifold Approximation and Projection (UMAP).** UMAP [17,48] is a method that projects high-dimension data into a lower-dimensional embedding and optimizes this representation to preserve the local and global structure of the data. The algorithm operates by generating a weighted graph of the databases on a nearest neighbor search, creating a “fuzzy” topological representation of the data. The lower-dimension embedding is then optimized through the minimization of cross-entropy to preserve the topological structure and relationships between data points. Key parameters like the number of nearest neighbors and the minimum distance between points allow users to balance the preservation of local versus global structures and control the clustering tightness. UMAP utilizes an approximate nearest neighbor search, providing scalability to work with large datasets. Van den Akker et al. [49] recently adapted a UMAP-based approach to characterize phenotypic patterns associated with NPM1 mutations using clinical flow cytometry data from patients with AML. As a feature extraction method, our group has shown that UMAP can be used, followed by discretization and classification by classical machine learning [26]. The advantage of this technique is near-linear run-times (i.e., a computational complexity of O(~n)) and is relatively robust against data shifts; however, its forward inference is relatively inefficient, and its memory requirements show non-linear increases, limiting the dataset sizes that can be used. In general, UMAP and its cousin t-SNE are better poised for data visualization rather than diagnostics given these issues.

**t-Distributed Stochastic Neighbor Embedding (t-SNE).** t-SNE is a statistical method for visualizing high-dimensional data that utilizes probability distributions in mapping to lower-dimensional space. This algorithm operates by finding the pairwise similarity between inputs and assigning higher probability to points that are close in proximity. The process aims to preserve these similarities in the lower dimension using a gradient descent to minimize the divergence in the distributions between the original and transformed feature space until a stable state of the embedding is achieved. While t-SNE excels at capturing local similarities, it struggles to accurately capture global relationships in the context of large datasets with many observations, e.g., events, in flow cytometry data. Belkina et al. [50] recently described “opt-SNE”, which automates perplexity and learning rate selection based on the dataset size, reducing the number of iterations needed for model convergence. This improved efficiency makes this method more efficient for large datasets, demonstrating high mapping quality in datasets with millions of cells; nevertheless, the fundamental t-SNE algorithm is still limited by higher computational complexity [O(n^2^)], and so the scaling of the datasets is non-linear. Interestingly, a recent innovation, flt-SNE [51], is promising in that it shows a purported complexity of O(n) by leveraging the very efficient fast Fourier transformation (FFT); however, we have not seen any published implementations using this technique.

## 5. Weakly Supervised Methods

While datasets have grown and methods have matured, the time required to review, annotate, and make sense of the large amount of data has remained fixed. Supervised methods offer strong precision and great results, but it can be laborious to build a high-quality labeled dataset in an efficient or consistent manner. Unsupervised methods offer some advantages, allowing the use of raw label-free data, but require review and interpretation to make use of the results. Given these constraints and limitations, weakly supervised and hybrid methods have been developed, often combining elements of the two in order to bridge the gap between making use of large datasets while still being able to leverage high-quality annotations.

A weakly supervised machine learning algorithm requires annotation labels only for a subset of the data or for aggregated data, as in the case of sample- or panel-level results in single-cell cytometry. Numerous methods have been developed to deal with the problem of weak supervision, many utilizing neural networks. These offer a flexibility that often allows for the superior matching of a model to the problem and constraints at hand.

While weakly supervised methods can offer improved performance due to the ability to scale to ever larger datasets, this can come at a significant cost. These types of models are sometimes large in scale, containing millions or even billions of parameters. To effectively train them, they require datasets that are equally sized to avoid overfitting, as well as more computational infrastructure and expertise. Additionally, more computational and software engineering experience is required to manage the additional complexity and possibility of distributed computing hardware involved. We, furthermore, recommend more effort be given during development in evaluating and understanding performance metrics, since biases may become harder to notice.

### 5.1. Common Methods Used in Weakly Supervised Models

**Semi-supervised learning**: Often, semi-supervised learning involves the use of pseudo-labeling, where, between training epochs, highly confident predictions are added to a set of augmented ground truths for the next epoch of training [52,53,54].

**Self-supervised learning (SSL)**: Self-supervised training is widely referenced in pre-training strategies and can involve teacher–student training or unsupervised techniques to build an embedding space out of a diverse set of data. Techniques such as BYOL, SimCLR, and DINOv2 highlight unique self-supervised techniques that exist in the computer vision space, which can be easily leveraged with single-cell data [54,55,56].

**Data augmentation**: A hallmark of both semi-supervised and supervised learning methods is the judicious use of data augmentation, either with noise, scaling/affine transformations, or other randomness, and censorship (masking) to help make use of unlabeled data or obtain the most leverage out of labeled examples.

**Multiple instance learning**: Multiple instance learning, referenced earlier, is a strong example of a type of weak supervision algorithm where an aggregation of one numerous modality, e.g., single-cell measurements, is used to train a label at a group label, e.g., cohort or specimen level. Multiple instance learning can be seen as a case of multimodal deep learning, with one modality being a set of data, and the other, labels at the specimen level.

In situations where an annotation is logically attached to a grouped item, such as a specimen which has many single-cell measurements that can be annotated, an aggregation method is required. Multiple instance learning extends the concept of a batch of data into a batch of batches, which are called “bags”. A bag is some selection of individual measurements which are combined with some aggregation like a mean or pooling operation. An aggregation can be simple or more complex such as an attention-based aggregation [33,57].

### 5.2. Representation Learning

**Contrastive learning:** Contrastive learning builds a representation space by computing a loss between items that are known to be different or similar. It is a form of metric learning, seeking to build a vector space representation with a meaningful measure of distance.

Some common techniques include *the Siamese or twin Network*, with two twin encoders with a “Siamese” loss comparison between the output (see Figure 3) [58]; *the Triplet Network*, which is an iteration on the idea of the Siamese network where three encodings are created with a loss that compares a single element to another that is similar and one that is dissimilar [59,60]; and *SimCLR*, which is a popular contrastive SSL algorithm in the computer vision literature that could be informative in the contrastive learning of single-cell data [61].

**Embeddings**: In general, the goal of representation learning is to seek to learn vector space embeddings that have valuable algebraic properties. This can be via simple “featurization” of inputs; however, embedding models continue to grow in complexity and utility.

Note that embeddings are a valuable way to enhance clustering algorithms. It is required for any machine learning use of data, but distance methods are common unsupervised methods. Nearest neighbor and many clustering methods can operate directly on “featurized” databases of embeddings. Embeddings can often be used as a dimension reduction technique, though it can be common to expand the dimension of the data as well [62,63].

### 5.3. Generative Models

**Auto-encoders and variational auto-encoders:** An auto-encoder (AE) is a type of neural network that encodes and decodes data, traditionally in a bottleneck pattern. It can be thought of as a neural network “identity” function. Variational inference applied to an auto-encoder, termed “Variation auto-encoder” or VAE, allows for a probabilistic interpretation of the generated latent embedding features. It can also serve as a type of anomaly detection by monitoring the reproduction error of data run through both the encoder and decoder [64,65,66,67,68,69]. The auto encoding structure is a common pattern seen in several successful neural networks in recent years [70,71]. Masked auto-encoders combining the censorship of input data with newer network architectures have shown great results in more recent research [72,73,74].

**Other generative models**: Any probabilistic model can be thought of as “generative” but, typically, the distribution of data to represent is complex. While VAEs are excellent at providing an intuitive probabilistic interpretation of an embedding space, their generative capacity lacks behind newer techniques. If one has goals to make use of generated single-cell data, more advanced generative models such as the Generative Adversarial Network (or one of its many variations) or ideas from stable diffusion may offer improved fidelity [75,76].

### 5.4. Foundation Models and Transfer Learning

Foundation models are large-effort machine learning projects that aim to be useable to a variety of downstream-related tasks. They often focus primarily on a model that provides abstract vector embedding output.

Foundational models are trained on networked clusters of accelerated compute hardware where one must consider how to effectively distribute data (as well as model parameters) across a network. Luckily, foundation modeling artifacts are easily shared and can be later tuned to run efficiently on commodity hardware. A single large foundation model can help serve individual researchers and teams for downstream tasks and use cases.

**Transformer architecture**: The “Transformer” is a neural network architecture, novel for its primary use of efficient *self-attention* blocks and its subsequent ability to scale to billions of trainable parameters [71]. Attention, in this sense, is an algorithmic mechanism hoping to mimic the cognitive ability to focus on salient and relevant attributes within a complicated context [77]. Though it was built and first used for the language domain, it has been shown to be highly performant in vision and other domains. While its structure of blocks is made up of an encoder–decoder architecture, it differs significantly from the idea of an AE- or VAE-type model [72,78,79].

**Pre-training and transfer learning**: In popular ML modalities, large pre-training on a diverse set of data has been shown to improve downstream models [80,81]. A usually large unsupervised dataset is used to develop a backbone (or foundational) model that outputs tensor embeddings and usually more complex neural network output (transformer self-attention blocks). Pre-training on large datasets can help in cases of “few-shot” identification or domain adaptation.

**Teacher–student training**: Bootstrap Your Own Latent (BYOL) is an unsupervised approach that utilizes augmentations of input data and exponential momentum updates to a target network [82,83].

While many of these weakly supervised techniques represent successful paradigms in other domains, their use in traditional flow cytometry is under-represented. Even though this is the case, understanding the techniques can allow them to be adapted to any dataset as appropriate, and represents new paths that a flow cytometry researcher could persue.

## 6. Tools and Infrastructure

### 6.1. Programming Languages

While most major programming languages offer frameworks and support for machine learning, the two most widely used are Python and R. Both have distinct advantages, and more than a few fiery discussions have been waged on internet forums regarding their suitability. Both languages are interpreted and dynamically typed, making them suitable for interactive environments such as Jupyter notebooks. Both offer an extensive universe of libraries, packages, and frameworks for almost any use case. R was originally developed for statistical programming as a free alternative to the commercial language S, and features excellent support for traditional statistics and data visualization. Python, in contrast, is a general-purpose programming language with very wide adoption not only in machine learning, but across many software engineering disciplines. Dominant deep learning frameworks such as PyTorch [84], Tensorflow [85], and JAX [86] are all Python-based. Python also offers a very shallow learning curve and is one of the first languages studied by computer science students. Perhaps the most significant downside to Python is its slow performance: pure Python code is many times slower than comparable code written in C, Rust, Go, or Java. For this reason, most Python-based deep learning frameworks are a mixture of Python and lower-level higher-performing C or C++ and hardware-specific (CUDA) code, which allows for easier access to the substantially greater computational efficiency from graphics processing units (GPUs).

While most machine learning projects are written in Python or R, a few notable alternatives exist. A worthy competitor here is Julia [87], responsible for the first two letters of the Jupyter notebook environment. Julia is a general-purpose interpreted language similar to Python but designed from the ground up for performance, numerical analysis, and visualization [88]. Julia is relatively new, with a version 1.0 release in 2018, and has since found a devoted following, although there are fewer libraries and frameworks available than exist for Python. A final project worth mentioning is Mojo, a new language still under development designed to address some of the performance and other issues that affect Python deep learning projects. Mojo is an “AI-first” language that aims for full compatibility with Python while offering additional extensions to boost performance and avoid the mixture of Python, C, and lower-level code that comprises most modern machine learning frameworks.

### 6.2. Models, Products, and Lifecycles

In academic settings, machine learning projects for flow cytometry often focus on model development. That is, the primary focus is to craft new algorithms or techniques that improve diagnostic accuracy or efficiency, with the end goal of publishing an academic paper. In a clinical setting, however, model development is the first step down a long road. All models require supporting software to execute correctly, store results, handle errors, provide logging, manage permissions and authentication, and display results. This supporting software is essential to successful clinical deployment and has a scope and complexity often greater than the model itself. Software development of this sort should be handled by professional software engineers, not data scientists, and will often require a product manager who can help understand and refine user requirements, coordinate work between scientists, engineers, and pathologists, and manage releases.

After models are deployed into the lab, Medical Directors and other laboratorians will inevitably seek to replace them with more accurate or capable successors. This continual deployment and validation of new models requires tools to manage and compare different model versions (see Figure 4). Managing multiple models across development, certification, and production environments is a complex task, and a new field of ML operations (MLOps) has arisen to address these unique complexities [89,90]. Sophisticated flow cytometry labs with multiple models, assays, and a full-time data science team will likely require dedicated MLOps personnel working in coordination with software engineers to manage the model lifecycle.

### 6.3. Infrastructure

Both the development and deployment of machine learning models requires a decision about how to run the software. A common decision point is whether to use cloud-based services or on-premises (“on-prem”) hardware. Successful products can be designed for either strategy, but an important additional consideration is the stage of the product lifecycle. The criteria that are important to optimize for research and model development are very different from the criteria that are important for clinical validation or production deployments. During initial model development, most data scientists will benefit from rapid iteration, ease of experimentation, and hands-on low-friction environments, while considerations such as robustness and scalability are secondary. In contrast, clinical production deployments require high uptime guarantees, reliability, near-complete automation, and thorough logging and monitoring, all features which typically add complexity and slow down iteration cycles.

Overall, cloud services are easy to scale, require little initial capital outlay, and can be made very resilient to outages, disasters, and other adverse events. However, storage costs can grow continually over time, and managing the resources can be complex and require dedicated professionals to configure and maintain the resources. Because of their low initial cost, cloud resources can be a wise choice for smaller labs or companies without a dedicated IT team and datacenter. For larger companies that operate their own datacenters, the cloud may be less appealing. Nevertheless, the scalability and robustness of cloud services make them a particularly good fit for production usage.

In contrast, on-prem hardware, either in a corporate datacenter or on local workstations, may be easier to access and utilize, making it a good fit for early-stage model development and research. While multiple cloud services offer AI workbench and hosted notebook products that may be appealing if local hardware is limited, these solutions may require more management and configuration than local solutions, and GPU time in the cloud can be expensive. For companies with sufficient resources, using local hardware for daily experimentation and development and cloud products for production can offer the best of both worlds.

## 7. Clinical Implementation

The clinical implementation of machine learning systems involves not only model training and development, but also deeper questions about software implementation, data security, and workflow design in addition to the usual considerations of clinical validation. Software development and infrastructure, as described above, is challenging, particularly in a clinical environment. From a regulatory perspective, guidance has been put out by many agencies on both sides of the Atlantic [91]; however, from a practical standpoint, the implementation of these methods by a clinical laboratory (LDTs) is a great regulatory unknown, as guidance is focused on commercial products pushed through the PMA or 510(k) pathway with the FDA.

Within the laboratory environment, validation encompasses two main tasks [26]: analytical evaluation of the model performance and a functional evaluation of the end-to-end (ETE) data pipeline, transformation, model prediction, and prediction serving. Analytical validation is a straightforward evaluation of the specificity and sensitivity of a model on a held-out dataset (see Section 2 above) and proceeds after model development, including the determination of a proper threshold to dichotomize the model output. Given the temporal variation in flow cytometry data, it is recommended that this validation set is temporally nearer to real-world clinical cases than the training set [92]. This is to gain better insight into the extent of “data shifts” that may have occurred after the model training was completed. Data shifts occur when real-world clinical data diverge from the distributions and correlations that were present in the training data. They, along with concept drift, are primarily responsible for the performance deterioration typically seen in machine learning applications over time and must be monitored for and addressed [93]. Given the fungible nature of data, it is possible to use post-hoc data temporally split for training and testing rather than waiting for new data to arrive for validation; however, the prospective analysis of performance is still recommended. After the model is validated and achieves the desired metric of goodness, the model can be converted into an independent free-standing module (e.g., a docker container), which allows for portability and adds robustness in the face of environmental variable changes (e.g., package updates or server patches).

Functional validation involves ETE testing from FCS file generation on the cytometer through resulting to either a technologist or pathologist depending on the model use case. These types of data validation tasks are particularly crucial as the laboratory information and electronic health systems within which these applications will be deployed are a complex ever-evolving ecosystem, which is often changing in ways that are unpredictable and difficult to plan for. While ETE testing can be laborious, including setting up a dedicated dummy environment complete with enough operational systems to make the test realistic, in our experience, ETE testing is the most robust method to set up these data pipelines. When used properly in close collaboration with the laboratory staff, this method can help uncover unusual and unique error modes that could not be detected otherwise. In a sense, ETE can serve as a wet dress rehearsal without interfering with busy clinical operations.

Given the regulatory and policy uncertainty regarding laboratory-developed tests, it is impossible to comment on what pathway would be required to satisfy the FDA’s final rule [94]. Likewise, as of November 2024, one of the FDA safe harbors, the New York State Department of Health Clinical Laboratory Evaluation Program, has not issued significant guidance regarding the use of the AI/ML tools, particularly regarding requirements for development, validation, and monitoring.

## 8. Potential Uses for Discovery

Classical supervised machine learning methods are typically applied to make predictions from a set of known entities using the labels provided in the training set. These methods are often very useful in finding complex combinations of features that maximize the differences between populations as defined by the labels we supply. However, these labels are often infallible, incomplete, and subject to the preconceived notions of the human factors and systems that provided them. Machine learning approaches can be extended beyond these narrow well-defined use cases into tools for discovery and hypothesis generation. This exploratory focus is particularly relevant to flow cytometry, where high-dimensional data and the sheer volume of information make hypothesis generation for capturing hidden patterns or rare cell populations a feasible endeavor.

One critical application of machine learning in discovery is the identification and exploration of rare or previously uncharacterized cell populations. Traditional manual gating strategies, though effective for predefined populations, often overlook subtle phenotypic variations with significant biological or clinical implications. As outlined in Section 4, clustering algorithms group cells based on phenotypic similarity, enabling the detection of these rare subpopulations. These methods are particularly effective for identifying disease-associated or immune-responsive cell populations. For example, FlowSOM has demonstrated utility in parsing high-dimensional cytometry data to uncover rare populations or detect minimal residual disease in acute myeloid leukemia (AML) [95]. The high dimensionality of flow cytometry data poses challenges for traditional analysis, as meaningful patterns are often obscured. Dimensionality reduction techniques (as discussed above) enable the distillation of complex data into interpretable forms. These techniques make it possible to visualize relationships among biomarkers and identify patterns in cell behavior. For example, UMAP, which excels at preserving both local and global data structures, has been used to identify phenotypic shifts associated with disease progression and treatment in leukemia patients [96]. These exploratory approaches have demonstrated their utility in the works of Hu et al. [31], who trained a deep learning model to predict latent cytomegalovirus infection and discovered a highly predictive population of CD8+, CD94+, and CD27- T lymphocytes in positive cases, and Evrard et al. [97], who analyzed populations of memory T cells to discover previously unappreciated heterogeneity at baseline and in response to inflammatory processes.

In addition to visualization, machine learning facilitates hypothesis generation through weakly supervised and hybrid approaches, which leverage minimal annotations or inferred labels. Multiple instance learning (MIL) is a prominent technique in this area, aggregating cell-level data to make specimen-level predictions without requiring exhaustive annotations for every cell. This method has been applied successfully to predict genetic aberrations in leukemia cases from flow cytometry data alone [35]. Self-supervised learning methods, such as contrastive learning, train models to recognize relationships within data without explicit labels, producing embeddings that can uncover new biological relationships. These embeddings, combined with clustering algorithms, have been used to hypothesize links between cellular signaling pathways and immune system responses in infectious disease models [98].

Another emerging frontier in discovery-driven machine learning is the integration of flow cytometry data with complementary modalities, such as next-generation sequencing, immunohistochemistry, and morphology-based diagnostics. Multimodal approaches provide a holistic view of biological systems by correlating phenotypic markers across several methodologies in the hopes that a more comprehensive evaluation of pathophysiology is achieved [99]. Recent advancements in transformer-based architectures and foundational models further enhance this capability, aligning diverse datasets into unified representations for analysis.

Despite its promise, discovery-driven machine learning presents unique challenges. One major limitation is the interpretability of complex models, particularly those based on deep learning. “Black-box” algorithms make it difficult to assign biological relevance to identified patterns. Efforts to address this include the development of explainability frameworks, such as SHAP (SHapley Additive exPlanations) [100] and attention mapping approaches, and the incorporation of domain knowledge into machine learning workflows. Scalability is another challenge, as high-throughput cytometry datasets necessitate robust computational resources. Cloud-based platforms and optimized algorithms have improved scalability, but the computational demands of high-dimensional analyses remain significant.

Bias and generalizability are additional concerns, as preprocessing steps and experimental variability can introduce biases that affect model performance. Models trained on specific datasets often fail to generalize to others due to batch effects or variability in experimental conditions. Techniques like transfer learning and domain adaptation offer potential solutions to these issues, allowing models to adapt to new data while maintaining performance. Finally, hypotheses generated through machine learning require rigorous validation using orthogonal methods or experimental studies. Collaboration between computational scientists, experimental biologists, hematopathologists, and clinicians is critical to ensure that ML-driven discoveries translate into actionable insights.

## 9. Conclusions

Machine learning methods have the potential to revolutionize diagnostics in clinical flow cytometry in terms of generating operational efficiencies, and, even more excitingly, in leading to new discoveries. It is obvious that the cost of developing infrastructure is non-trivial, requiring teams of data scientists and software engineers and for many institutions; this may be prohibitive. Nevertheless, the potential operational efficiencies have driven intense interest in ML not only in flow cytometry, but clinical medicine and the world economy as a whole. Interestingly, while most academic interest is focused on ML algorithms as they apply to flow cytometry, in our experience, much of our efforts are devoted towards building the infrastructure to support a clinical-grade ML pipeline, the documentation of model development, and the evaluation for clinical validation, and the major considerations required in operationalizing these tools. Indeed, the dearth of data, experience, and practitioners is a major hinderance [26,92], and we hope that efforts in implementing these solutions into clinical workflows will continue to make their way into the literature, regardless of their ultimate success. Likewise, while feature discovery can be empowered by ML, by implementing these models and building out the infrastructure required to operationalize a pipeline, major synergies can be found with regard to the quality and quantity of data that can be used. Currently, most of the literature focused on feature discovery is geared towards smaller datasets (dozens to hundreds) [95,96,98], whereas clinically accessible data can be easily 1–2 orders of magnitude greater (thousands to tens of thousands of patients) [25,26,29], opening possibilities for rare diseases and diseases with weak features. With the decreasing cost of storage and computation, tackling these datasets will inevitably become easier with time, opening these possibilities for scientific discovery to a variety of centers if and only if they make the investments in operational infrastructure to support ML implementation. With these investments, we are hopeful that we will begin to see a virtuous cycle where ML investments will lead to operational efficiencies that, in turn, foster additional resources for novel feature discoveries that will serve as the basis of future assay designs.

## Figures and Tables

**Figure 1 cancers-17-00483-f001:**
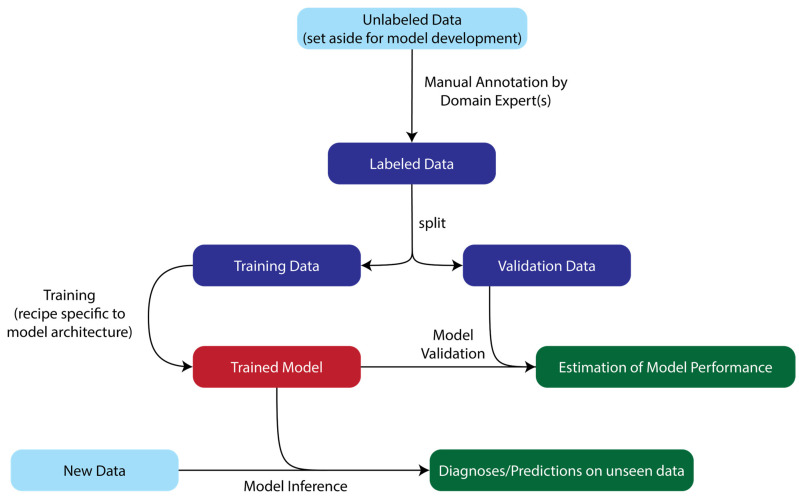
A diagrammatic overview of the model development workflow for supervised machine learning, from data annotation through training and validation to production-ready inference.

**Figure 2 cancers-17-00483-f002:**
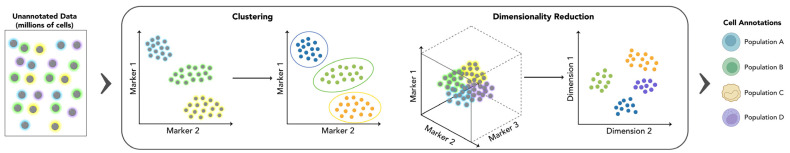
Unsupervised learning algorithms. Labelled cells with unknown identities are analyzed by flow cytometry and partitioned in the feature space based on marker expression. Clustering methods are used to group data points based on their marker profiles, while dimensionality reduction can be applied to high-parameter datasets to simplify their representation and enhance the biological interpretability. These tools output cell annotations to identify distinct populations, enabling the characterization of disease states and facilitating novel discoveries.

**Figure 3 cancers-17-00483-f003:**
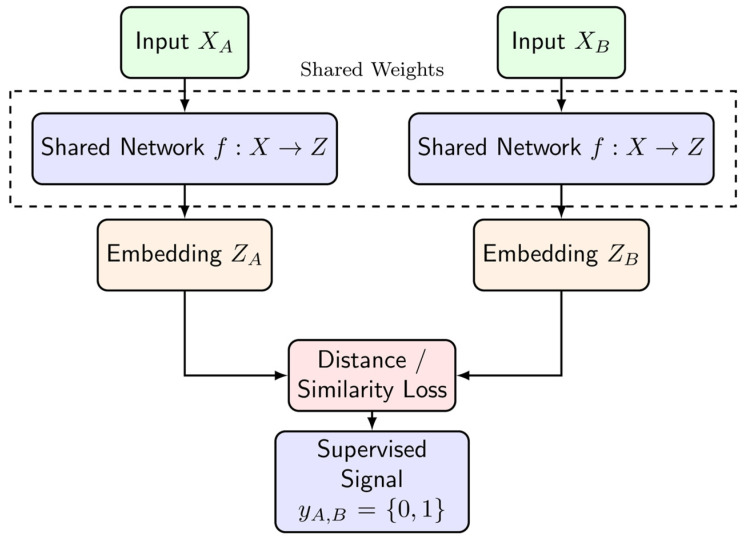
Twin networks: one neural network model is learned on paired examples. Pairs are weakly supervised, only requiring a label for if they are the “same” or not. N.b.: this supervised signal could be developed via an unsupervised method.

**Figure 4 cancers-17-00483-f004:**
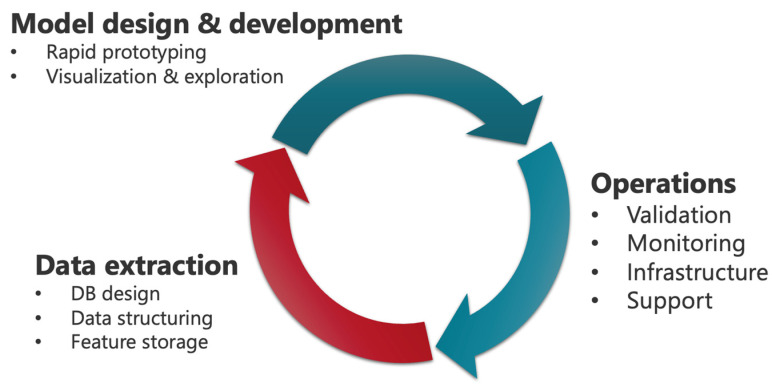
AI model lifecycle from initial data analysis to model development and operationalization.

**Table 1 cancers-17-00483-t001:** A summary of common machine learning algorithms with their advantages, disadvantages, and common uses.

Algorithm	Class	Description	Advantages	Disadvantages	Common Uses
Support Vector Machines (SVMs)	Supervised	Finds a decision boundary (hyperplane) that maximizes the margin between classes (for classification) or fits the best line/hyperplane (for regression). Uses kernels to handle non-linear boundaries.	Effective in high-dimensional spaces Robust against overfitting with regularization Able to handle complex boundaries	Computationally expensive Relatively uninterpretable	General classification/regression
Decision Trees	Supervised	Uses a tree-like model of decisions based on feature values. Each internal node represents a test on a feature; each branch is an outcome of the test, and each leaf is a class/regression outcome.	Easy to interpret and visualize Handles numerical and categorical data Fast training and inference	Prone to overfitting if not pruned Limited in complexity Susceptible to class imbalance	Simple systems Applications requiring interpretability
Random Forest	Supervised	An ensemble of decision trees, aggregated by voting for classification or averaging for regression. Each tree is trained on a bootstrapped subset of data with random subsets of features.	Minimal hyperparameter optimization Robust against outliers and noise Handles high-dimensional data	Less interpretable than a single tree Resource-intensive to train Typically less effective than boosting	General purpose model for tabular data
Gradient-Boosted Trees (XGBoost)	Supervised	Sequentially builds an ensemble of weak prediction trees, where each subsequent tree attempts to correct the errors of the previous ones. XGBoost is a popular optimized framework for gradient-boosting.	State of the art for tabular data Handles missing data and outliers well Highly tunable for performance nuance	Prone to overfitting Hyperparameter tuning is expensive Requires careful regularization	Highly effective models for tabular data
Neural Networks	Variable	Inspired by the structure of biological neurons. Consists of layers of interconnected “neurons” that learn hierarchical representations of data through backpropagation.	Captures complex non-linear relationships Highly flexible architectures Scales well with large datasets	Computationally intensive Requires large training datasets Hyperparameter tuning is complex, but crucial	Effective models for tabular, language, vision, and more
K-means Clustering	Unsupervised	Groups data into KK clusters by minimizing within-cluster variance. Iteratively updates cluster centroids and assignments until convergence.	Simple to implement Fast for moderate-sized datasets Requires spherical well-separated clusters	Must specify the number of clusters Sensitive to outliers Poor performance on varying cluster layouts	Customer segmentation Image compression Data exploration
Density-Based Spatial Clustering of Applications with Noise (DBSCAN)	Unsupervised	Groups together points that are closely packed together (points with many nearby neighbors), marking as outliers the points that lie alone in low-density regions.	Can find arbitrarily shaped clusters Robust against outliers/noise	Poor for clusters with varying densities Sensitive to hyperparameter choices	Geospatial data analysis Anomaly detection Clustering with irregular shapes/densities
Gaussian Mixture Models (GMMs)	Unsupervised	Assumes data are generated from a mixture of a finite number of Gaussian distributions with unknown parameters. Each Gaussian distribution is characterized by its mean and covariance.	Probabilistic cluster membership Can model overlapping clusters Flexible to distribution type	Must specify the number of components Sensitive to initialization Can converge to local optima	Probabilistic clustering Anomaly detection Data distribution modeling
Principal Component Analysis (PCA)	Unsupervised	Transforms data into new orthogonal axes (principal components) that capture the directions of maximum variance. The top components retain most of the variance in the data.	Effective dimensionality reduction Speeds up subsequent training Removes correlation among features	Fails to capture non-linear relationships Principal components lack interpretability	Dimensionality reduction Preprocessing for other models
Non-Negative Matrix Factorization (NMF)	Unsupervised	Factorizes a non-negative data matrix into the product of two smaller non-negative matrices, interpreting data as “parts-based” additive combinations.	Produces interpretable decomposition Allows text and image input	Sensitive to initialization and local minima Only applicable to non-negative data May require careful tuning	Topic modeling in text analysis Image feature extraction Recommender systems
Self-Organizing Maps (SOMs)	Unsupervised	Neural network that uses competitive learning. Maps high-dimensional data onto a low-dimensional (usually 2D) grid, preserving topological or neighborhood structure in the data.	Good for dimensionality reduction Preserves topological relationships Can reveal cluster structures visually	Susceptible to hyperparameters Tougher to interpret than linear methods Computationally intensive	High-dimensional data visualization Exploratory data analysis
Uniform Manifold Approximation and Projection (UMAP)	Unsupervised	Graph-based method for dimensionality reduction that approximates the manifold structure of the data. Preserves local and global structure, often used for visualization in 2D or 3D.	Preserves local and global data structure Fast, even on large datasets Produces visually interpretable embeddings	Embeddings are sensitive to hyperparameters Interpretation of axes is not straightforward Non-deterministic unless forced	High-dimensional data visualization Exploratory data analysis
t-Distributed Stochastic Neighbor Embedding (t-SNE)	Unsupervised	Non-linear technique that converts distances between points into probabilities, aiming to preserve local neighborhoods in a lower-dimensional space (usually 2D).	Imparts clusters on high-dimensionality data	Computationally intensive Requires hyperparameter tuning Potential for misleading visual artifacts	High-dimensional data visualization Exploratory data analysis
Auto-encoders	Unsupervised	Neural networks that learn to compress (encode) data into a latent space and then reconstruct (decode) it. Can be adapted for denoising, anomaly detection, or generative tasks.	Learns complex non-linear embeddings Can reduce dimensionality and remove noise Highly versatile	Prone to overfitting Architecture has significant performance impact Difficult to interpret	Dimensionality reduction Noise reduction Synthetic data generation
Transformers	Variable	Neural network architecture that uses self-attention mechanisms to process input sequences in parallel, avoiding recurrence. Originally designed for NLP but extended to images and more.	State of the art in many tasks Highly parallelizable training Can capture long-range dependencies	Data- and compute-intensive training Larger models are slower at inference time Simpler models may be sufficient for simpler tasks	Large language models Foundation models

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
