# Peer review of "Machine Learning Methods in Clinical Flow Cytometry"

_cancers, 2025, doi:10.3390/cancers17030483_

Round 1

Reviewer 1 Report

Comments and Suggestions for Authors

Comments on the manuscript identified as cancers-3373927, in which 3 learning methods are reviewed to facilitate the analysis of hematological neoplasia data by flow cytometry as a clinical interpretation tool. Some comments are listed below.

The background of the study is consistent with the title of the study, however, a serious comment is to review the paragraphs that lack scientific support and recommend adding references.

On the other hand, it is recommended to include the meaning of MODEL DEV in the caption of figure 1.

Include a general conclusion of the study.

Author Response

Comments on the manuscript identified as cancers-3373927, in which 3 learning methods are reviewed to facilitate the analysis of hematological neoplasia data by flow cytometry as a clinical interpretation tool. Some comments are listed below.

The background of the study is consistent with the title of the study, however, a serious comment is to review the paragraphs that lack scientific support and recommend adding references.

On the other hand, it is recommended to include the meaning of MODEL DEV in the caption of figure 1.

Response: Thank you for this suggestion, we have replaced the abbreviated “dev” with “development”.

Include a general conclusion of the study.

Response: Thank you for this suggestion, we have a conclusion in the new Section 9. It highlights the key takeaways and future directions of the chapter, and is reproduced below.

“Machine learning methods have the potential to revolutionize diagnostics in clinical flow cytometry in terms of generating operational efficiencies but, even more excitingly, in leading to new discoveries. It is obvious that the cost of developing infrastructure is non-trivial, requiring teams of data scientists and software engineers and for many institutions, this may be prohibitive. Nevertheless, the potential operational efficiencies have driven intense interest in ML not only in flow cytometry, but clinical medicine and the world economy as a whole. Interestingly, while most academic interest is focused on ML algorithms as they apply to flow cytometry, in our experience, much of our efforts are devoted towards building the infrastructure to support a clinical grade ML pipeline, documentation of model development and evaluation for clinical validation, and the major considerations required in operationalizing these tools.  Indeed, the dearth of data, experience, and practitioners is a major hinderance and we hope that efforts in implementing these solutions into clinical workflows will continue to make their way into the literature, regardless of their ultimate success. Likewise, while feature discovery can be empowered by ML, by implementing these models and building out the infrastructure required to operationalize a pipeline, major synergies can be had with regard to the quality and quantity of data that can be used. Currently most literature focused on feature discovery is geared towards smaller datasets (dozens to hundreds)88–90 whereas clinically accessible data can be easily 1-2 orders of magnitude greater (thousands to tens of thousands of patients) opening possibilities for rare diseases, and diseases with weak features. With the decreasing cost of storage and computation, tackling these datasets will inevitably become easier with time opening these possibilities for scientific discovery to a variety of centers if and only if they make the investments in operational infrastructure to support ML implementation.  With these investments we are hopeful that we will begin to see a virtuous cycle where ML investments will lead to operational efficiencies that in turn foster additional resources for novel feature discoveries that will serve as the basis of future assay designs.”

Reviewer 2 Report

Comments and Suggestions for Authors

Thanks for your work. Before acceptance, some revisions should be made for clearer presentation to the reader. Specific suggestions are listed below.

(1) An overall logic diagram, reflecting the membership relationships between the described different AI methods, should be provided.

(2) An overall table should be provided, including the basic information of each AI method such as the concise definitions, key features, main advantages, and suitable conditions.

(3) The focus of this paper is an review of the application of machine learning in clinical flow cytometry. However, there is a lot of content to describe the basic principles of each machine learning method in Section 3 and its following Sections, and less review on the existing application of machine learning in clinical flow cytometry. For those machine learning methods that have been applied to this research topic, the author should review in detail and clarify the strengths and weaknesses of these methods in this research topic. For those machine learning methods that have not yet been applied, the author should indicate potential application directions in this research topic. In this way, it is better to inspire readers.

Author Response

Thanks for your work. Before acceptance, some revisions should be made for clearer presentation to the reader. Specific suggestions are listed below.

(1) An overall logic diagram, reflecting the membership relationships between the described different AI methods, should be provided.

(2) An overall table should be provided, including the basic information of each AI method such as the concise definitions, key features, main advantages, and suitable conditions.

Response: Thank you for this valuable feedback. We agree that a clarifying element was missing and have added a new Table 1 that combines both of these suggestions. It summarizes the approaches discussed in the text, provides a short description of their approach, advantages, disadvantages, and common uses, and highlights the relative group memberships across the algorithms. An example row is reproduced below.

Gradient-Boosted Trees
(XGBoost)

Supervised

Sequentially builds an ensemble of weak prediction trees, where each subsequent tree attempts to correct the errors of the previous ones. XGBoost is a popular, optimized framework for gradient-boosting.

State-of-the-art for tabular data
Handles missing data and outliers well
Highly tunable for performance nuance

Prone to overfitting
Hyperparameter tuning is expensive
 Requires careful regularization

Highly effective models for tabular data

(3) The focus of this paper is an review of the application of machine learning in clinical flow cytometry. However, there is a lot of content to describe the basic principles of each machine learning method in Section 3 and its following Sections, and less review on the existing application of machine learning in clinical flow cytometry.

For those machine learning methods that have been applied to this research topic, the author should review in detail and clarify the strengths and weaknesses of these methods in this research topic.

For those machine learning methods that have not yet been applied, the author should indicate potential application directions in this research topic. In this way, it is better to inspire readers.

Response: We agree that these details would significantly improve the work. To this end, we have added brief summaries of the advantages and disadvantages of the respective methods throughout the chapter. Additionally, to make it easier for the reader to place these approaches in context, we have added a new Table 1, that aggregates this information in one place. For those methods that have not yet been applied, we have added our conjectures as to likely areas of application in the near future.

Reviewer 3 Report

Comments and Suggestions for Authors

Flow cytometry is widely used to analyze blood cell subpopulations to manage clinical situations such as hematological malignancies or immunological dysfunction. While its early use involved only a few markers, thus allowing data analysis by experienced physicians, the steady increase of the number of studied markers generated as well as the remarkable progress of artificial intelligence make it an important and timely challenge to process clinical data with optimal machine learning methods. The submitted review is therefore of high interest for the medical community. This reviewer found the manuscript well written and highly informative. He strongly recommends publication.

However, there are a few points of concern that the authors might want to consider to improve the manuscript.

while the description of machine learning methods was generally found very insightful and informative, some minor points were felt difficult to understand :

- line 137: is overfitting the use of an excessively complex model or the excessive training of a given model (early stopping might be mentioned somewhere).

- line 231 : how does the loss function reflect a "tradeoff" - The tradeoff is rather the choice to accept a relatively high loss value ?

- line 339 : perhaps it might be useful to add that clustering is dependent on the definition of distance - i.e. parameter scaling, and acknowledge that there may not be any golden standard since cell populations are difficult to define.

- line  537: it is certainly useful to mention the transformer architecture. But it is difficult to understand the basic principle on reading the decription given by the authors. How is attention mimicked by the algorithmic treatment ?

- line 579: while this is a minor point, the GPU capacity to dramatically increase processing velocity might be mentioned in line with "CUDA".

- line 607: the importance of data validation might deserve some more emphasis, since this is an essential requirement to incoroporate a new procedure into standard medical practice. Thus, defining the kappa index might provide a simple way of explaining why the simple consideration of "prediction accuracy" might be insufficient.

- Perhaps the term of "data shift" might be mentioned together with a specific medical reference (e.g. Lea et al., New England J Med 390:293-295., 1024).

The reviewer felt that the description of the relevance of flow cytometry to medical practice was less detailed than that of artificial intelligence methods. Thus, the immunological interest of analysing lymphocyte subpopulations might be supported by some specific emphasis and references (e.g. Hu et al., PNAS 117:21373–21380, 2020 ; Evrard et al., Immunity 56, 1664–1680, 2023) in addition to e.g. reference 40.

Note a minor typo in the summary :
- line 8 : "to effectively in analyzing"

Author Response

Flow cytometry is widely used to analyze blood cell subpopulations to manage clinical situations such as hematological malignancies or immunological dysfunction. While its early use involved only a few markers, thus allowing data analysis by experienced physicians, the steady increase of the number of studied markers generated as well as the remarkable progress of artificial intelligence make it an important and timely challenge to process clinical data with optimal machine learning methods. The submitted review is therefore of high interest for the medical community. This reviewer found the manuscript well written and highly informative. He strongly recommends publication.

However, there are a few points of concern that the authors might want to consider to improve the manuscript.

while the description of machine learning methods was generally found very insightful and informative, some minor points were felt difficult to understand :

- line 137: is overfitting the use of an excessively complex model or the excessive training of a given model (early stopping might be mentioned somewhere).

Response: Thank you for this insightful feedback. We have adjusted this paragraph to clarify the key points and added an explicit mention of early stopping in the techniques sentence. The updated version is reproduced below.

“Achieving an optimal balance ensures that a model generalizes well to new, unseen data by being neither overly simplistic (known as underfitting) nor excessively complex (known as overfitting). High bias results in models may consistently miss relevant patterns, for example, collapsing to a single pattern found in annotated labels and ignoring rare labels, while high variance leads to models tailored too closely to the training data and unable to perform accurately on testing data. Avoiding under- and overfitting is a crucial consideration throughout the model development process, with implications for training data inclusion/exclusion or transformation, model architecture selection, and hyperparameter tuning, among others. Techniques such as cross-validation, regularization, early-stopping, and careful model selection are often employed to mitigate these risks, ensuring the model performs robustly across different datasets.”

- line 231 : how does the loss function reflect a "tradeoff" - The tradeoff is rather the choice to accept a relatively high loss value ?

Response: Thank you for highlighting this source of confusion. We have added clarity to this trade-off by rephrasing the paragraph to read as follows:

“The optimization problem is solved subject to a loss function which is a sum of two terms, one which seeks smoothness of the decision boundary (which should help generalization) and the other which seeks to minimize the misclassification of training examples (since an excessively smooth boundary may miss important features of the data). Minimizing the total loss function reflects a tradeoff between the competing demands of these smoothness and accuracy terms, with their relative importance usually controlled by a tunable hyperparameter.”

- line 339 : perhaps it might be useful to add that clustering is dependent on the definition of distance - i.e. parameter scaling, and acknowledge that there may not be any golden standard since cell populations are difficult to define.

Response: We thank the reviewer for bringing this oversight to our attention and agree this is a valuable point to highlight.   We have added the following discussion on distance metrics in flow cytometry:

“We note that performance of clustering methods in flow cytometry can vary greatly depending on how similarity between cells is measured. Calculating feature similarity is heavily influenced by the distance metric chosen and how each parameter (e.g., intensity, scatter) is weighted, which can dramatically change clustering results. For instance, Euclidian distance exhibits high granularity and is ideal when absolute marker intensities are imperative while cosine similarity prioritizes the orientation of data points and is adept at capturing relative expression patterns to distinguish phenotypes. Further, transformations (i.e. log, arcsinh) and scaling methods- commonly employed to optimize performance across a wide dynamic range- can modify the data distribution, thereby altering the distance between points and biasing the definition of each subpopulation. Further, cell populations tend to overlap and may not fall into discrete clusters in a biological context. Consequently, there is no gold standard for defining populations in unsupervised methods and gating strategies can vary between experts, instruments, or experiments. Accordingly, the best approach varies by dataset and objective where empirical testing is needed to determine the appropriate distance metric to apply.”

- line  537: it is certainly useful to mention the transformer architecture. But it is difficult to understand the basic principle on reading the description given by the authors. How is attention mimicked by the algorithmic treatment ?

Response: We have rephrased the introduction to transformer architectures to focus more on clarirty. The updated version is reproduced below:

“The “Transformer” is a neural network architecture, novel for its primary use of efficient self-attention blocks and its subsequent ability to scale to billions of trainable parameters.  Attention, in this sense, is an algorithmic mechanism hoping to mimic the cognitive ability to focus on salient and relevant attributes within a complicated context.”

- line 579: while this is a minor point, the GPU capacity to dramatically increase processing velocity might be mentioned in line with "CUDA".

Response: We have added an explicit mention of the increased efficiency gained through GPU-specific implementations, reproduced below:

“For this reason, most python-based deep learning frameworks are a mixture of python and lower level, higher performing C or C++ and hardware-specific (CUDA) code, which allows for easier access to the substantially greater computational efficiency from graphics processing units (GPUs).” 

- line 607: the importance of data validation might deserve some more emphasis, since this is an essential requirement to incorporate a new procedure into standard medical practice. Thus, defining the kappa index might provide a simple way of explaining why the simple consideration of "prediction accuracy" might be insufficient.

Response: Thank you for pointing out this opportunity to better contextualize the clinical implementation work needed for these approaches. We have added the following line emphasizing the importance of data validation.

“These types of data validation tasks are particularly crucial, as the laboratory information and electronic health systems within which these applications will be deployed are a complex, ever-evolving ecosystem which is often changing in ways that are unpredictable and difficult to plan for.”

- Perhaps the term of "data shift" might be mentioned together with a specific medical reference (e.g. Lea et al., New England J Med 390:293-295., 1024).

Response: We agree, and have added the following lines:

“This is to gain better insight into the extent of “data shifts” that may have occurred after the model training was complete. Data shifts occur when real-world clinical data diverges from the distributions and correlations that were present in the training data. They, along with concept drift, are primarily responsible for the performance deterioration typically seen in machine learning applications over time and must be monitored for and addressed.”

The reviewer felt that the description of the relevance of flow cytometry to medical practice was less detailed than that of artificial intelligence methods. Thus, the immunological interest of analysing lymphocyte subpopulations might be supported by some specific emphasis and references (e.g. Hu et al., PNAS 117:21373–21380, 2020 ; Evrard et al., Immunity 56, 1664–1680, 2023) in addition to e.g. reference 40.

Response: Thank you for this feedback. We agree that the added context provided by these examples would help readers understand and apply these approaches to their own work. We have added explicit mention and concise summaries of these prior works in the discovery paragraph, reproduced below.

“One critical application of machine learning in discovery is the identification and exploration of rare or previously uncharacterized cell populations. Traditional manual gating strategies, though effective for predefined populations, often overlook subtle phenotypic variations with significant biological or clinical implications. As outlined in section 4, clustering algorithms group cells based on phenotypic similarity, enabling the detection of these rare subpopulations. These methods are particularly effective for identifying disease-associated or immune-responsive cell populations. .... These exploratory approaches have demonstrated their utility in the works of Hu et al., who trained a deep learning model to predict latent cytomegalovirus infection, and discovered a highly predictive population of CD8+, CD94+, CD27- T lymphocytes in positive cases, and Evrard et al., who analyzed populations of memory T cells to discover previously unappreciated heterogeneity at baseline and in response to inflammatory processes.”

Note a minor typo in the summary :
- line 8 : "to effectively in analyzing"

Response: Corrected. Thank you.

Round 2

Reviewer 2 Report

Comments and Suggestions for Authors

This paper in current version can be accepted.